



# Nonlinear processes in tsunami simulations for the Peruvian coast with focus on Lima/Callao

Alexey Androsov[1], Sven Harig[1], Natalia Zamora[2], Kim Knauer[3], and Natalja Rakowsky[1]

[1]Alfred Wegener Institute for Polar and Marine Research (AWI), Bremenhaven, Germany
[2]Barcelona Supercomputing Center (BSC), Spain
[3]EOMAP GmbH  Co. KG (EOMAP)

**Correspondence:** Alexey Androsov (alexey.androsov@awi.de)

**Abstract.**

This investigation addresses the tsunami flooding in Lima and Callao caused by the massive 1746 earthquake (Mw 9.0) along the Peruvian coast. Numerical modelling of the tsunami flooding processes in the nearshore includes strong nonlinear numerical terms. In a comparative analysis of the calculation of the tsunami wave effect, two numerical codes are used, Tsunami-HySEA and TsunAWI, which both solve the shallow water (SW) equations but with different spatial approximations. The comparison primarily evaluates the flow velocity fields in flooded areas. The relative importance of the various parts of the SW equations is determined, focusing on the nonlinear terms. Particular attention is paid to the contribution of momentum advection, bottom friction, and volume conservation. The influence of the nonlinearity on the degree and volume of flooding, flow velocity and small-scale fluctuations is determined. The sensitivity of the solution with respect to the value of the bottom friction parameter is investigated as well.

## 1 Introduction

Over the past couple of decades, after the catastrophic 2004 Mw 9.3 Sumatra earthquake and widespread tsunami (Titov et al., 2005a; Kowalik et al., 2005; Grilli et al., 2007; Syamsidik et al., 2021), there has been an increased need for a more detailed study of these natural disasters. Their understanding has greatly improved due to the development of numerical models and methods that can reproduce various hazard and risk scenarios and use them for community preparedness and mitigation (Berke and Smith, 2009; Shmueli et al., 2021). The tuning required to reduce numerical uncertainties beyond those already inherent in natural phenomena can be resolved by comparing the effects of different numerical schemes.

According to their purpose, tsunami models can be conditionally divided into two types. The first type includes operational models (Titov et al., 2005b), whose main task is to estimate the time of arrival of a tsunami wave and its height on a time scale faster than wave propagation in real-time. Often such models are linearized and do not perform calculations in the flood zone. The risk of flooding can be assessed using the so-called "amplification factor", which describes the relationship between the wave height in the open sea and the maximum flood height for waves with different wave characteristics (Glimsdal et al., 2019). Another approach to assessing the tsunami threat can be considered based on Green's summation, using the parameters of the





seismic source (Miranda et al., 2014) as input. Green's functions are calculated using a numerical SW model for linearized
equations at points of most significant interest.

Numerical models of this kind should describe the flood zone with a very high spatial resolution and have reliable numerical flooding/drainage schemes, which is associated with relatively high energy consumption in the computational aspect. In addition, often, the time between the occurrence of a tsunami and its approach to the coast is minimal, and then pre-created databases of possible scenarios of tsunami sources and numerical modelling (Macías et al., 2017; Rakowsky et al., 2013) come
to the rescue. The advantage of such models compared to operational ones is a more accurate and detailed description of the processes occurring in the flood zone (Baba et al., 2014; Harig et al., 2022). The choice of operational information is no longer based on calculations during the event and analysis of flood zones according to some assumptions but on the appropriate scenario, which can be refined when correcting the source of tsunami wave formation (Harig et al., 2020).

Numerical models for calculating surface gravity waves, including tsunami waves, within the framework of SW theory,
depending on various approximations, can be divided into linear, nonlinear, and nonlinear-dispersive. In operative models, linear equations are often used to reduce computational costs (Babeyko et al., 2010). Another advantage of such models is associated with a significantly higher time step of integrating the problem since there is no limitation on the advective mode (Androsov et al., 2002).

Another class of equations contains three types of nonlinearity - momentum advection, nonlinearity in the continuity equa-
tion due to variable water thickness, and nonlinear friction (Androsov et al., 2011; Macías et al., 2017). All these types of nonlinearity play different roles in wave propagation near and on the coast. As shown in the work of Androsov et al. (2013), on a steep slope of the bottom when approaching the coast, the momentum advection plays a prominent role; in the shelf zones, there is already a nonlinearity in the continuity equation and, near the coast and on it, friction at the bottom (Ribal, 2008).

Many papers deal with the analysis of nonlinearity in tsunami models. The main emphasis in these works is on comparing the
wave amplitude in linear and nonlinear problems in general. Some link analytical analysis with practical calculations (Zahibo et al., 2006) and others compare solutions for linear problems with nonlinear systems for some areas of the World Ocean affected by tsunami waves (Pujiraharjo and Hosoyamada, 2009; Liu et al., 2009; Saito et al., 2014). In this case, the effect of this or that kind of nonlinearity is not singled out. At the same time, the analysis of each of the components of the nonlinearity is highly dependent on the details of the local conditions, determines the importance of the relevant factors for each component
of the nonlinearity and can help in choosing the appropriate model in a given situation.

The nonlinear-dispersion terms qualitatively and quantitatively change the amplitude and shape of the wave as it passes over an underwater obstacle or wave runup on a vertical barrier (Beji and Battjes, 1993; Viotti et al., 2014) and with an increase in the steepness of the tsunami wave front (Tsuji et al., 1991; Elsheikh et al., 2022). Accounting for dispersion in the model is associated with significant numerical limitations (ultra-high grid resolution and small integration step) and is mainly used
in comparative analysis (Horrillo et al., 2006; Pujiraharjo and Hosoyamada, 2009). Horrillo et al. (2006) noted that including dispersion in numerical models is necessary for a more accurate description of the interaction of a tsunami wave with partial reflection from the continental shelf and when entering sea bays where resonant oscillations can occur. However, there needs to be more coastal observations to support this. As for the passage of a tsunami wave in the deep ocean, a comparative analysis





shows that the nonlinear shallow water models are quite reliable for practical purposes and show a high degree of agreement
with observations, and the inclusion of dispersion only marginally improves the result (Pujiraharjo and Hosoyamada, 2009). In
this regard, we do not consider the dispersion terms in our study.

This work has a twofold purpose. The first relates to a comparative analysis of two numerical tsunami models, TsunAWI and
Tsunami-HySEA, using the example of a flood assessment caused by a strong destructive earthquake (Mw 9.0) that occurred
on October 28, 1746, and resulted in a wave of about 24 m in the city of Callao (Jimenez et al., 2013). Since the wave run-up
on the coast and its flooding is a highly nonlinear process, the second goal of the article is to analyze the role of each term
of the nonlinearity in the numerical solution. A comparative analysis of the space-time dynamics and energy characteristics
of various implementations of the TsunAWI model with various combinations of included nonlinear terms in the equations is
carried out.

The article briefly describes two tsunami models, TsunAWI and Tsunami-HySEA, based on nonlinear SW equations. A
distinctive feature of implementing these two models is their spatial discretization. TsunAWI operates on unstructured meshes
and solves the equations with the finite element method, while the Tsunami-HySEA uses structured nested meshes and employs
the finite volume method. The next section of the work is devoted to setting up the problem for these two models. Section 3
describes the simulation domain, initial conditions, and meshes characteristics. The calculation and comparative analysis of the
simulation results of the two models are given in Section 4. In Section 5, based on the calculations of the TsunAWI model, an
analysis of the nonlinearity contained in the equations is performed. Section 6 discusses the results of this work and concludes.
The Appendix provides extended comparison results of the two models.

## 2   Description of the models

The shallow water equations derived from vertically integrating the Reynolds-averaged Navier-Stokes equations under the
hydrostatic assumption and Boussinesq approximation in the $\Omega$ plane domain bounded by boundary $\partial\Omega$ are considered with
the general formulation as follows:

$$\partial_t \mathbf{u} + (\mathbf{u} \cdot \nabla)\mathbf{u} + f\mathbf{k} \times \mathbf{u} + g\nabla\zeta - \nabla \cdot A_h \nabla \mathbf{u} + \frac{gn^2\mathbf{u}|\mathbf{u}|}{H^{4/3}} = 0 \tag{1}$$

$$\partial_t \zeta + \nabla \cdot (\mathbf{u}H) = 0 \tag{2}$$

for the horizontal velocity vector $\mathbf{u} = (u, v)$ and the total water depth $H = h + \zeta > 0$, $h$ is the unperturbed water depth, and
$\zeta$ the surface elevation, $\nabla = (\partial/\partial x, \partial/\partial y)$ is the gradient operator, $f$ the Coriolis parameter, $\mathbf{k}$ the unit vector in the vertical
direction, $g$ the gravitational acceleration and $A_h$ the eddy viscosity coefficient. Estimating the bottom friction terms involves
an empirical friction Manning-Stricler parameter $n$, which depends on the bottom type and generally varies from 0 (friction-less
bottoms) to 0.06, with typical values in the range of 0.02 to 0.03 (Harig et al., 2008).





On the solid part of the boundary, $\partial\Omega_1$, and on its open part, $\partial\Omega_2$, we impose the following boundary conditions:

$$\mathbf{u}_n|_{\partial\Omega_1} = 0, \ \mathbf{\Gamma}(\mathbf{u},\zeta)|_{\partial\Omega_2} = \mathbf{F}, \tag{3}$$

where $\mathbf{u}_n$ is the velocity normal to the solid boundary, $\mathbf{\Gamma}$ is the operator of the boundary conditions and $\mathbf{F}$ is the known vector-function determined by the boundary regime (Oliger and Sundström, 1978; Androsov et al., 1995) and different for inflow and outflow. In tsunami models, the most common boundary information is the radiation boundary condition: $\mathbf{u}_n = \sqrt{gH^{-1}}\zeta$ are used.

The problem (1-3) for $\mathbf{v} = (\mathbf{u},\zeta)$ is solved for given initial conditions:

$$\mathbf{v}|_{t=0} = \mathbf{v}^0. \tag{4}$$

The equation for the energy of the external motion mode, whose equations are vertically averaged equations, is obtained by multiplying the first of equations (1) by $\rho_0 H u$, the second by $\rho_0 H v$, and equation (2) by $\rho_0(g\zeta + \frac{1}{2}|\mathbf{u}|^2)$. The total energy of the mean motion, in this case, will be determined by the formula:

$$E = \frac{1}{2}\rho_0(H|\mathbf{u}|^2 + g\zeta^2), \tag{5}$$

is the total energy per unit area, where $\rho_0$ is the average density of the seawater. The first term of the right hand is kinetic energy, and the second is the potential energy of the flow.

## 2.1   TsunAWI model

The main reason for choosing the finite element method is the computational grid, which can be adapted to cover basins with
uneven bottom topography and coastline without generating nested meshes. The finite element spatial discretization in the TsunAWI model is based on the approach by Hanert et al. (2005) with some modifications like added viscous and bottom friction terms, corrected momentum advection terms, radiation boundary condition, and nodal lumping of a mass matrix in the continuity equation. The basic principles of discretization follow Hanert et al. (2005) with linear conforming elements $P_1$ for sea surface height $\zeta$ and water depth $h$ and linear non-conforming elements $P_1^{NC}$ for the velocity $\mathbf{u}$. The basic principles of
the finite element discretization follow the paper of Hanert et al. (2005), and we are not repeated here.

Simulation of tsunami wave propagation benefits from using an explicit time discretization. Numerical accuracy requires relatively small time steps, reducing implicit schemes' main advantage. Furthermore, modelling the inundation processes usually requires very high spatial resolution in coastal regions (up to meters) and, consequently, many nodes, drastically increasing necessary computational resources in case of implicit temporal discretization. The leap-frog scheme was chosen as
a simple and easy-to-implement method. We rewrite equations (1) and (2) in time-discrete form:

$$\frac{\mathbf{u}^{k+1} - \bar{\mathbf{u}}^{k-1*}}{2\Delta t} + (\mathbf{u}^k \cdot \nabla)\mathbf{u}^k + f\mathbf{k} \times \mathbf{u}^k + g\nabla\zeta^k -$$
$$-\nabla \cdot A_h \nabla\mathbf{u}^k + \frac{gn^2\mathbf{u}^{k+1}|\mathbf{u}^k|}{H^{4/3}} = 0,$$

$$\frac{\zeta^{k+1} - \bar{\zeta}^{k-1*}}{2\Delta t} + \nabla \cdot (H^k\mathbf{u}^k) = 0.$$





Here $\Delta t$ is the time step length, and $k$ is the time index. The leap-frog three-time-level scheme provides second-order accuracy and is neutral within the stability range. This scheme, however, has a numerical mode removed by the Robert-Asselin time filter procedure: $\bar{\mathbf{v}}^{k^*} = \mathbf{v}^k + \alpha(\mathbf{v}^{k+1} - 2\mathbf{v}^k + \bar{\mathbf{v}}^{k-1^*})$, where asterisks denote filtered characteristics and $\alpha = 0.01$ (Asselin (1972)).

Advection is one of the essential complexities in calculating the waves of a tsunami in a shallow zone. Advection of momentum in the original formulation by Hanert et al. (2005) is very unstable. For this reason, we applied a new approach to calculate advection. This approach has appeared robust in areas of complex morphometry where advection becomes significant. To calculate the advection term in the momentum equation, we first project the velocity from the $P_1^{NC}$ to the $P_1$ space to smooth it. Then we use it in the advection term and proceed by multiplying this form with a $P_1^{NC}$ basis function and integrating it over the domain. In contrast to the advection scheme proposed by Hanert et al. (2005), it has the advantage that no boundary integral has to be computed and is more stable.

For modelling wetting and drying, we use a moving boundary technique which utilizes extrapolation through the wet-dry boundary and into the dry region (Lynett et al., 2002). This concept excludes dry nodes from the solution and then extrapolates elevation and barotropic gradient to the dry nodes from their wet neighbours.

Because the leap-frog scheme is neutrally stable, it demands horizontal viscosity in places of large gradients. The horizontal viscosity coefficient is determined by a Smagorinsky parameterization (Smagorinsky, 1963) with adding some small background coefficient.

## 2.2 Tsunami-HySEA model

Tsunami-HySEA solves the two-dimensional shallow-water system using a high-order finite volume method. These methods are mass preserving for arbitrary (nested) bathymetries. Tsunami-HySEA implements several reconstruction operators:

- MUSCL (see van Leer, 1979), which achieves second order;

- the hyperbolic Marquina's reconstruction (see Marquina, 1994), which achieves third order;

- the TVD combination of piecewise parabolic and linear 2D reconstructions that also achieves third order.

For large computational domains and in the framework of TEWS, Tsunami-HySEA also implements a two-step scheme similar to leap-frog for the deep water propagation step and a second-order TVD-WAF flux-limiter scheme for close-to-coast propagation/inundation step. Combining both schemes guarantees mass conservation in the complete domain and prevents the generation of spurious high-frequency oscillations near discontinuities generated by leap-frog type schemes. At the same time, this numerical scheme reduces computational times compared with other numerical schemes while the amplitude of the first tsunami wave is preserved. Concerning the inundation modelling, the wet-dry fronts discretization consists of locally replacing the 1D Riemann solver used during the propagation step with another 1D Riemann solver considering the presence of a dry cell. This reconstruction step is also modified to preserve the water depth's positivity.



# 3 Models setup

## 3.1 Domain


The Lima/Callao region is located on the central coast of Peru, characterized by a narrow strip of land between the Pacific Ocean and the Andes. The coastal zone is relatively flat, with sandy beaches and rocky cliffs interspersed along the coastline.

The bathymetry of the area is diverse: shallow water near the coast gradually deepens with distance from the coast. The continental shelf is relatively narrow, and the water depth beyond it increases rapidly. The maximum depth in this area is about
2 km, located several hundred kilometres from the coast.

## 3.2 Bathymetry and topography data

The bottom relief is crucial for numerical simulations of wave propagation and inundation. In the deep ocean, we use the GEBCO dataset (General Bathymetric Chart of the Ocean version 2021) at a resolution of 15 arc seconds. EOMAP processed a fused bathymetric and topographic dataset for the nearshore range based on different sources. For the water area down
to about 300 m water depth, a combination of nautical charts and GEBCO was incorporated. Tandem-X topography data (Krieger et al., 2007) was used for the land area. Several minor errors and inaccuracies were removed and partially interpolated from the Tandem-X data. The final fused bathymetric and topographic dataset was derived in 1 m resolution and shown in Fig. reffig:LimaBathyTopo.

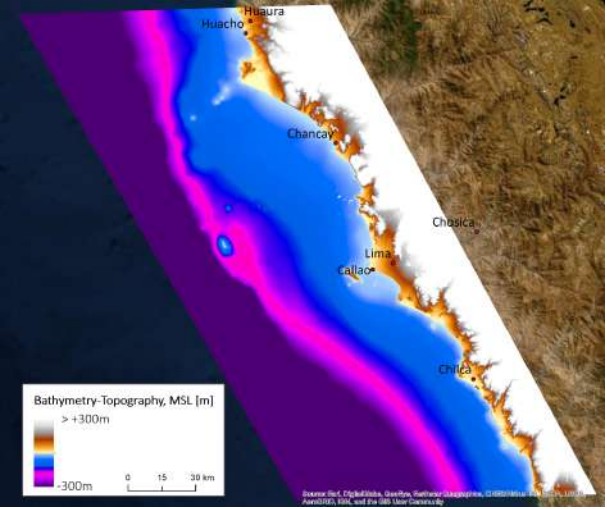

**Figure 1.** Section of the model domain with data processing by EOMAP.

## 3.3 Meshes

These numerical codes are based on two different meshes. In the case of Tsunami-HySEA, four levels of nested grids are used, and for TsunAWI, triangular meshes are required. To have similar resolutions along all domains between the numerical





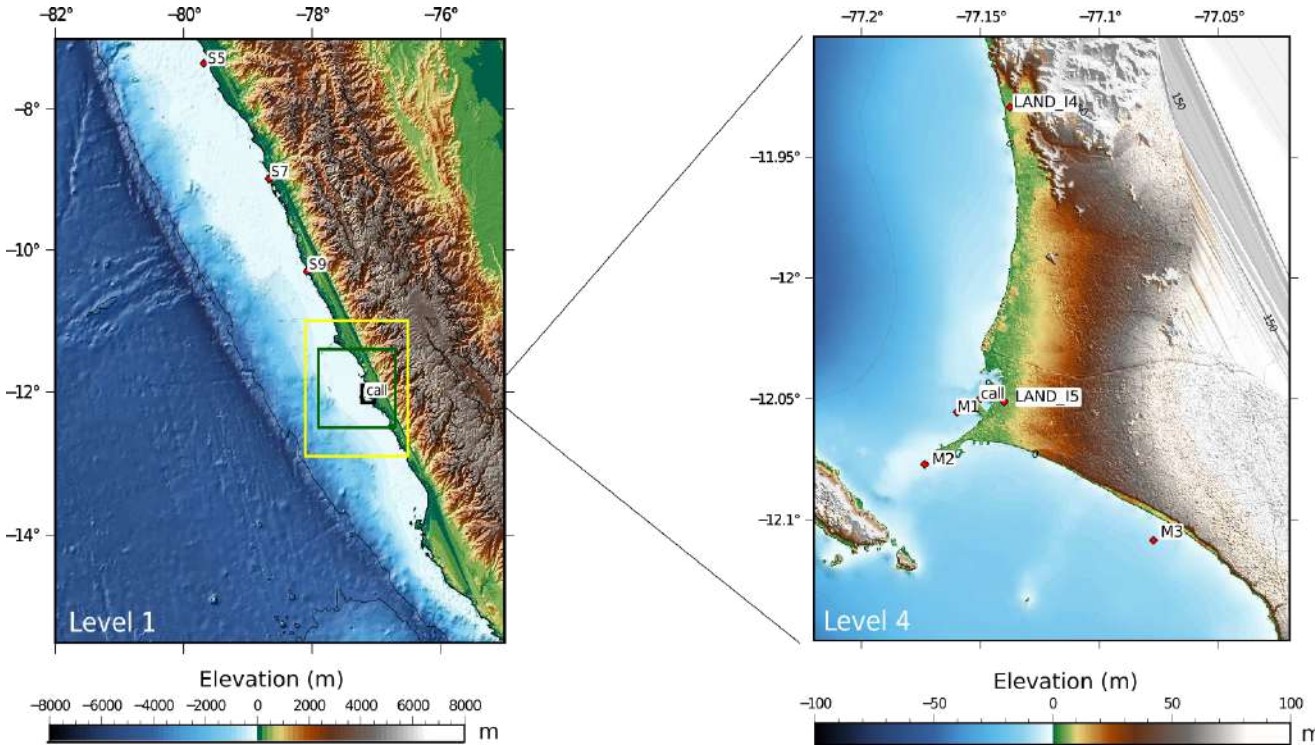

**Figure 2.** Topography and bathymetry grids used for Tsunami-HySEA numerical simulations. a) Larger domain (level 1) is used in the simulation, and four levels of the nested grids are shown with rectangles: level 2 (yellow), level 3 (green) and level 4 (black). b) Grid level 4 with the highest spatial resolution (12 m). $S5 - 9$ stands for stations located around 10 m depth, and $M1, M2$, and $M3$ are the same virtual buoys used in (Jimenez et al., 2013), and 'call' refers to the buoy location in Callao, La Punta according to the Intergovernmental Oceanographic Commission - IOC. Digital elevation model from Compilation (2019) and EOMAP.

simulations that are compared, the triangular meshes are created from the nested grids with resolutions from 60 arc seconds (∼1.85 km) to 12 m (Level 4). These meshes are shown in Fig. 2 and Fig. 3. The mesh resolution for TsunAWI is based on a criterion considering both water depth and bathymetry slope. A small section of the triangulation is displayed in Fig. 3.



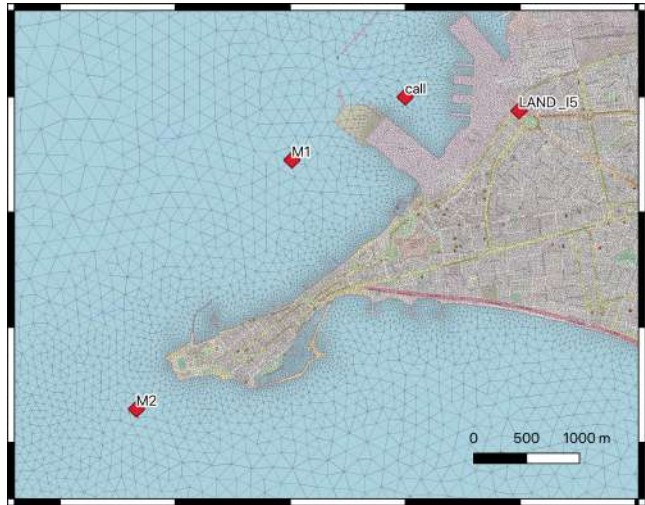

**Figure 3.** Small section of the triangular mesh used for TsunAWI simulations. The edge length of triangles ranges from 10 km in the deep ocean (beyond the scope of this picture) to about 10 m in the highest resolved part. Also included are some of the locations used to analyse the time series. Basemap: © OpenStreetMap contributors 2021. Distributed under the Open Data Commons Open Database License (ODbL) v1.0.

### 3.4 Model initialization

The event under consideration is the historic tsunami on 28 October 1746. We use the source parameters proposed in (Jimenez et al., 2013) with slight depth adaptation for the centre of the subfaults needed to generate initial conditions based on Okada analytical formulations (Okada, 1985). The depth of each subfault was extracted based on the model Slab2 (Hayes et al., 2018). That study includes simulations of the tsunami inundation as well. We compare the outcome of our modelling approach to their results, although bathymetry, topography data, and mesh resolution differ. Therefore complete agreement cannot be expected.

The source is shown in Fig. 4; it consists of five sub-faults, each with lengths of 140 km and widths of 40 km and with the given slip distribution with the moment magnitude Mw 9.0. An adaptation of the given source was needed to use the centre of each sub-fault, estimated in this case at 24 km. Although that paper considers a dynamic source, we restrict our investigations to a static source for the sake of simplified consistency in the setups of the numerical models. More details on the source definitions are contained in Jimenez et al. (2013).




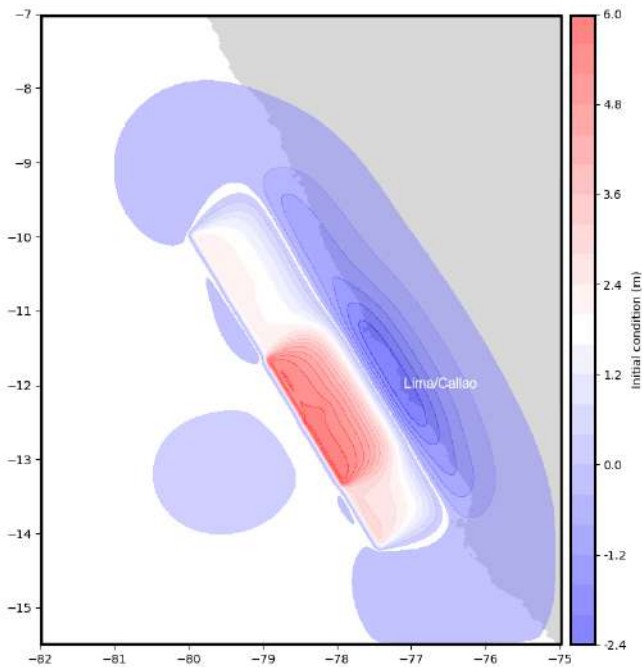

**Figure 4.** Initial conditions for the tsunami numerical simulations. Parameters modified from Jimenez et al. (2013).

Starting from this initial sea surface elevation and zero velocity, we simulate the tsunami propagation and inundation for four hours in real time. Besides the resulting inundation in Lima/Callao region, we investigate and compare tide gauge records in virtual and real offshore positions and the temporal evolution in selected land points.

## 4  Comparison of model results

This section compares the simulation results of tsunami wave propagation and run-up modelling for the two numerical models presented above. The spatial and temporal fields show a detailed comparison of the simulation results. Some of the comparison results are given in the Appendix.

The Tsunami-HySEA and TsunAWI models are initialized with the same height fields and integrated over a four-hour time interval. The energy distribution indicated by the maximum wave amplitudes received during this period is shown in Fig. 5.

### 4.1  Inundation area

The historical event 1746 resulted in considerable inundation in the Lima/Callao region (Jimenez et al., 2013). The flow depth in a part of the inundated area in the Callao region obtained by Tsunami-HySEA and TsunAWI is shown in Fig. 6. The extent corresponds qualitatively well with the result shown in Jimenez et al. (2013).



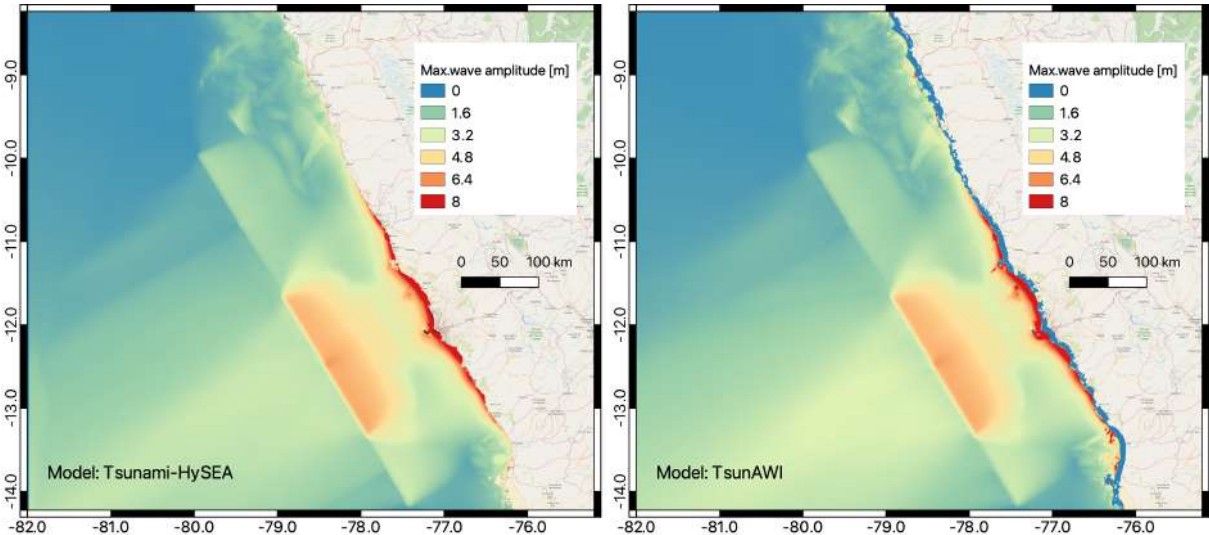

**Figure 5.** Maximum wave amplitude for a simulation of four hours propagation time in the models Tsunami-HySEA (left panel) and TsunAWI (right panel). Basemap: © OpenStreetMap contributors 2021. Distributed under the Open Data Commons Open Database License (ODbL) v1.0.

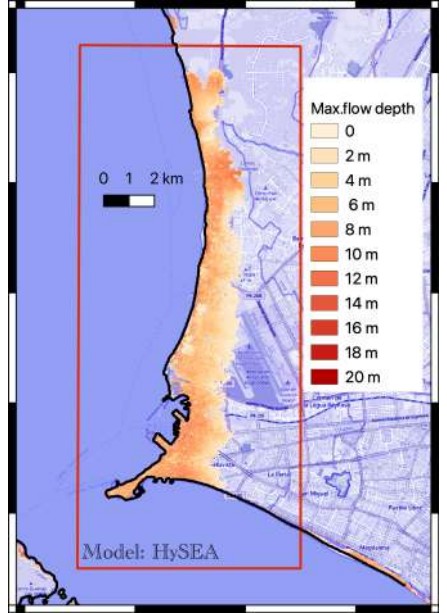
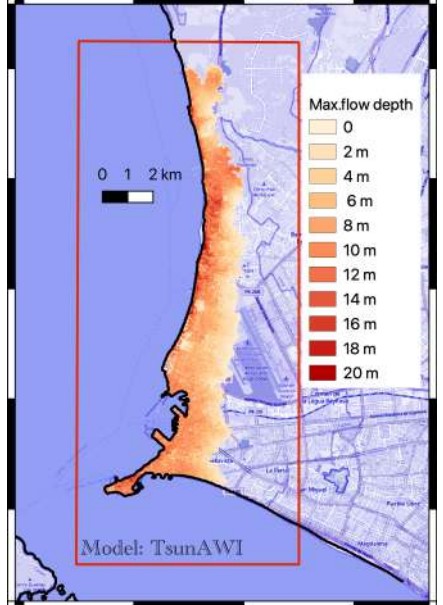

**Figure 6.** Maximum flow depth in the inundated area in Callao obtained by Tsunami-HySEA (left panel) and TsunAWI (right panel) for a Manning value of $n = 0.015$ and a simulation time of four hours. The box indicates the area in which the inundation extent and inundation properties were calculated. Basemap: © OpenStreetMap contributors 2021. Distributed under the Open Data Commons Open Database License (ODbL) v1.0.





**Figure 7.** Upper panels: Inundation extent for the smallest ($n = 0.015$) and largest ($n = 0.06$) Manning values used in this study obtained by Tsunami-HySEA (left side) and TsunAWI (right side). Lower panels: Functional relationship of Inundation area (in [km$^2$], bottom left) and volume (in [Mio. m$^3$], bottom right) for the full range of Manning values used in this study. Linear and quadratic regressions are shown. Basemap: © OpenStreetMap contributors 2021. Distributed under the Open Data Commons Open Database License (ODbL) v1.0.

The inundation extent, as well as the height distribution, depends on the bottom friction parameter in the inundated area.
Both models use bottom friction parameterisation in Manning form with a constant Manning coefficient throughout the domain. Results for Manning values in the range between $n = 0.015$ and $n = 0.06$ are shown in Fig. 7. The inundation area (with a minimum flow depth of 1cm) ranges from 21.85 km$^2$ ($n = 0.06$) to 30.03 km$^2$ ($n = 0.015$).

The lower panels of Fig. 7 depict the regressions for inundation area (in [km$^2$]) and volume (in [Mio. m$^3$]) concerning the Manning coefficient for both models. We consider the inundated land area in the box indicated in Fig. 6 and a minimum flow
depth of 1 cm. The volume is estimated by integrating the maximum flow depth over the entire inundation area. The functional dependency for both models is quite similar, with gradual differences. Especially the volume obtained by TsunAWI for small





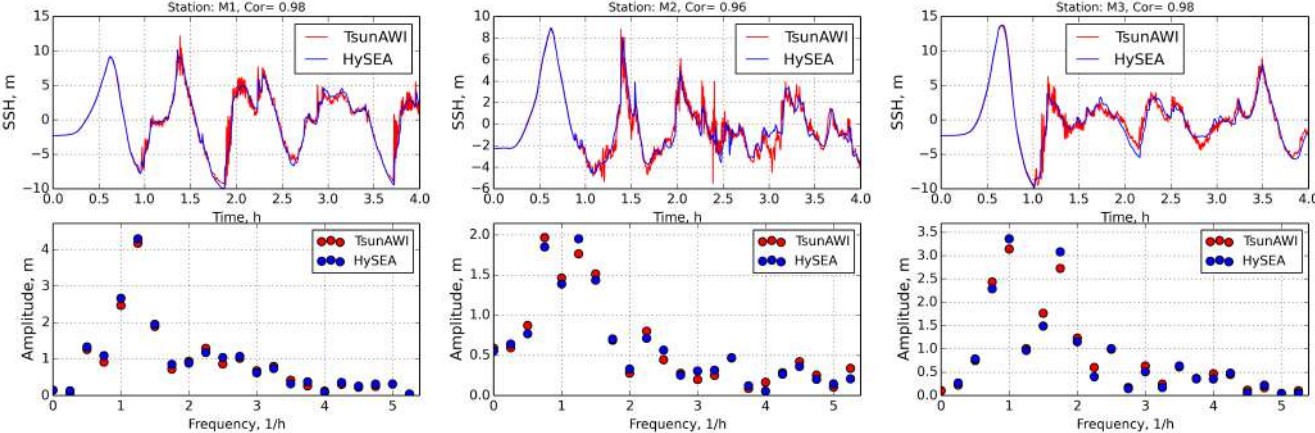

**Figure 8.** Wave amplitudes and their spectra for $M1 - M3$ stations in the implementation of two models Tsunami-HySEA and TsunAWI.

Manning numbers is considerably more significant than the one determined by Tsunami-HySEA since the numerical scheme in TsunAWI tends to larger values of the wave height, especially in the nearshore range (see also Fig. A2 in the Appendix). Over the full range of Manning values, we observe a reduction of the inundation area by 22.9% for Tsunami-HySEA and 27.2% for TsunAWI. The Appendix summarises More quantities in Table A1.

## 4.2 Time series comparison

We will continue the analysis of the results of the implementations of the two models by comparing the course of the wave height (see Fig. 8) and the horizontal velocity components (see Fig. 9) for the entire simulation period for stations $M1 - M3$. A comparison of the tsunami wave amplitudes shows a very high agreement with a correlation coefficient exceeding 0.96 for the two models. Note that the TsunAWI model shows somewhat greater instability in the solution. We can explain this because the Tsunami-HySEA model has higher stability in the frontal zones due to the stabilization of advection by schemes with flux-limiters. A more interesting fact is a slight shift in the maximum amplitudes at the frequencies of the primary tsunami wave (close to one hour). This is especially noticeable at stations $M2$ and $M3$, which are the most distant from the initial disturbance source. So, for example, at station $M3$, the amplitude of the hourly frequency dominates in the Tsunami-HySEA model, and at frequencies close to 50 minutes, the amplitude in the HySEA model somewhat dominates. The picture is similar for the wave with frequencies of $\sim 37$ and $\sim 42$ minutes, where the amplitude maximum changes in one model or another. We also note that as we move away from the initial source of perturbation, there is a greater discrepancy between the two solutions at the time of the onset of the wave maximum. Some additional experiments show that the finite element model has a slightly higher phase propagation velocity than the finite volume model for long waves. This effect was also mentioned in the article by Maßmann et al. (2010) when comparing the tidal problem in the North Sea for different spatial approaches.





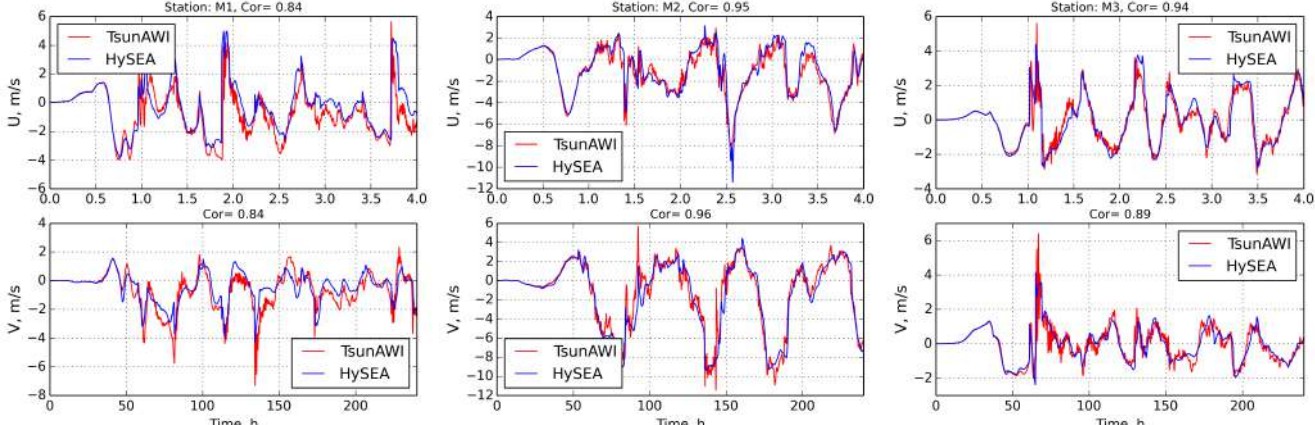

**Figure 9.** Comparison of horizontal velocity components at three stations $M1 - M3$ for two models Tsunami-HySEA and TsunAWI. The top panel is the u component; the bottom is the v component.

Comparing the horizontal velocity components shown in Fig. 9 at the same three stations show a high degree of similarity over the entire simulation period. The correlation coefficient of the two solutions at stations $M1 - M3$ is 0.84, 0.95 and 0.94, respectively.

The comparison of the two models presented in this section shows a high degree of agreement between the two solutions, which in turn allows us to choose the solution of TsunAWI for a more detailed analysis of the influence of nonlinearity on the behaviour of the tsunami wave in the coastal zone and the flood zone.

## 5 Nonlinearity in the TsunAWI simulations

The system of equations (1-2) contains three nonlinear terms: momentum advection, sea surface oscillations in the continuity equation, and nonlinear bottom friction. The analysis of the results of comparing the two models presented in the previous part of the work shows only minor differences in the redistribution of amplitudes at the main frequencies of the tsunami wave. And this difference becomes noticeable only at the stations at the most remote distance from the source of the tsunami wave. This allows us to analyze the influence of nonlinear terms on the solution using the results of one (TsunAWI) numerical model.

We performed four main nonlinear experiments on the propagation of a tsunami wave in the region of interest. The first experiment is the solution to the complete equations with all nonlinearity (full tsunami model: FTM). The second experiment (without momentum advection: WMA) is connected with disabling advection in the equations of motion, the second term in equations 1. In the third experiment, in continuity equation 2, the influence of the level fluctuation on the total layer thickness (without nonlinearity in continuity: WNC) was disabled in the velocity divergence term. Note that in the flood zone, as in all cases, we retain the total layer thickness $h + \zeta$. Finally, in the fourth experiment (only bottom friction: OBF), the nonlinearity in the equations is represented only by the bottom friction terms.





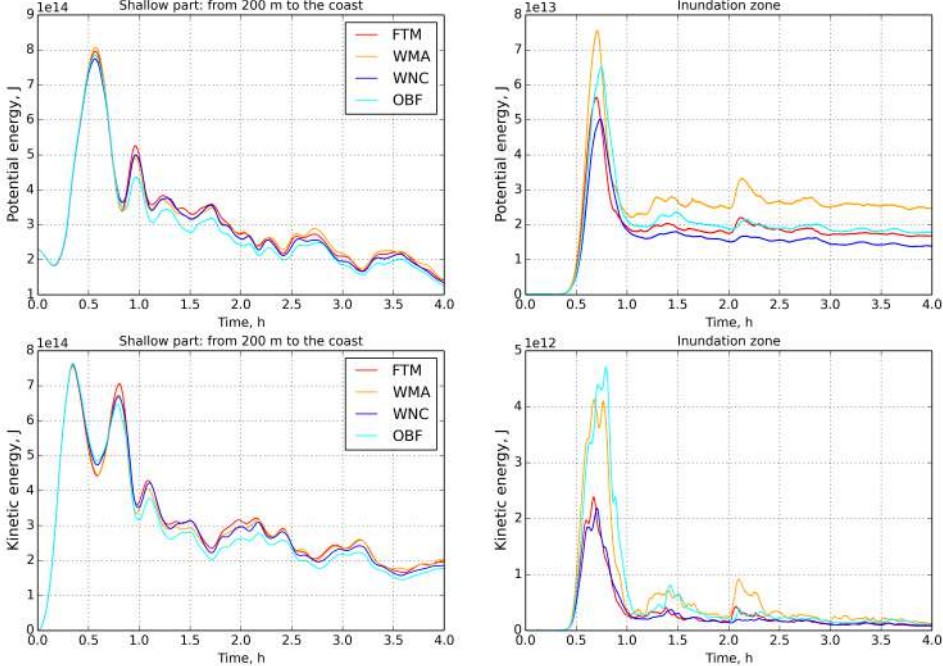

**Figure 10.** Potential and kinetic energy for shelf zone and inundation area for four experiments. Upper panel - potential energy; bottom panel - kinetic energy; left - shelf zone (from 200 m to coast); right - inundation area.

In the conclusion of this chapter, using various coefficients of bottom friction, an analysis is made of the influence of nonlinear friction on the solution in the inundation zone.

We begin our analysis of the influence of nonlinear terms on tsunami wave propagation by comparing the potential and kinetic energies (Eq. 5) of the four experiments described above. The computational domain was divided into three subdomains: deep water ($H > 200$ m), shelf (from 200 m to the coastline) and, in fact, the flood zone. For each of these areas, the potential

and kinetic energy is calculated. Figure 10 shows the values of the terms of the total energy equation for the shelf zone and the flooded area. We note that in the deep-water part of the region, the potential and kinetic energies have a practically insignificant difference in all four experiments. The difference between the terms of the energy equation in the shelf zone also turns out to be insignificant (Fig. 10, left panel) for the first incoming tsunami wave. Some visible differences in amplitudes appear for secondary waves and waves reflected from the coastline.

Figure 10 (right panels) shows the values of potential and kinetic energy only in the area of flooding. For this zone, the differences in the energy components are already quite significant. Thus, the potential energy of the WMA calculation exceeds the energy of the complete system of equations (FTM) by more than 35%. Calculating equations without non-linearity in the equation of continuity (WNC) shows an underestimation of the energy. In addition, an essential element of comparison is that in the WNC experiment, there is a slight phase shift of the onset of the maximum compared to the FTM experiment.





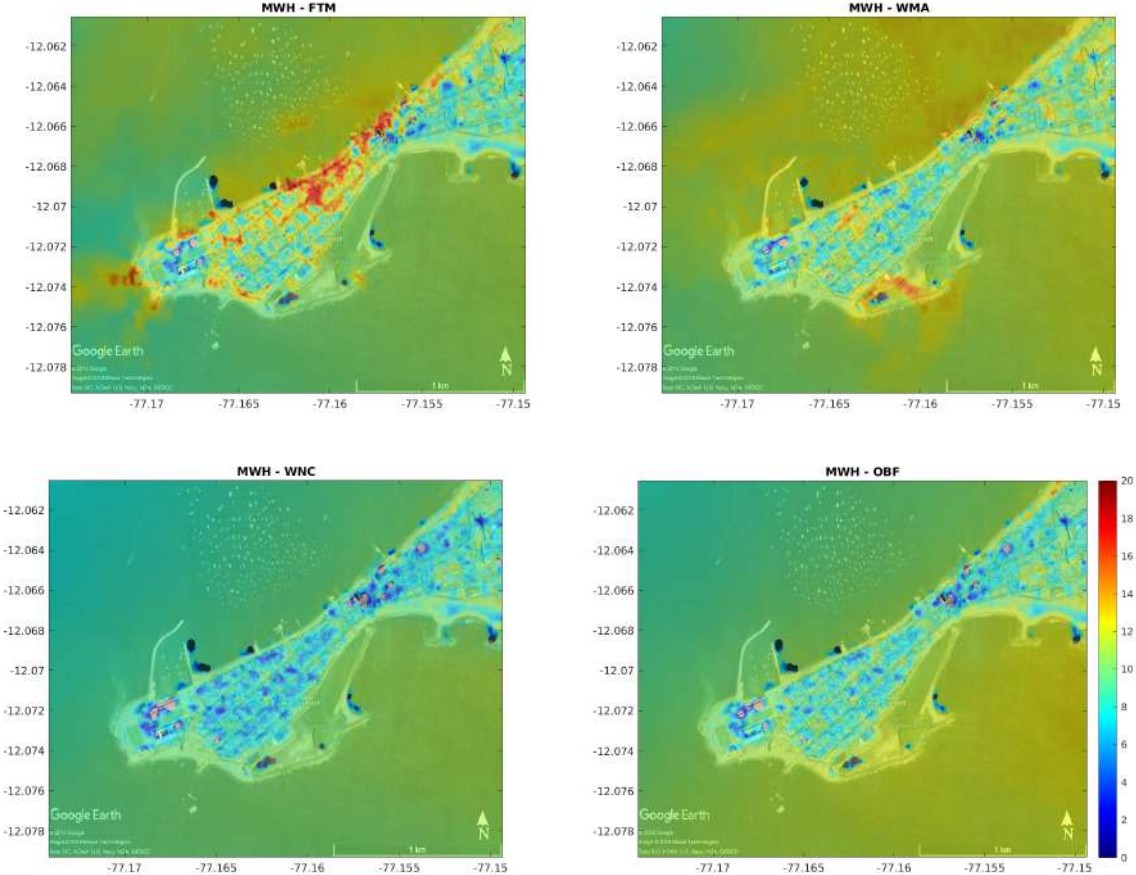

**Figure 11.** Maximum sea surface amplitude in [m] for the complete simulation period. Upper left panel: Full model (FTM); upper right: Without momentum advection (WMA); lower left: Omitting nonlinearity in the continuity equation (WNC); lower right: Only bottom friction (OBF). Basemap: © Google Earth 2018.

The OBF experiment contains the general tendencies of the WMA and WNC experiments - an overestimated amplitude and a characteristic shift of the maximum of the first wave.

The difference in kinetic energy in the coastal zone shows an even greater contrast. The absence of momentum in the equation of motion plays the leading role in kinetic and potential energy. The difference in amplitude compared to the basic experiment (FTM) reaches 50%. At the same time, we note that the calculation with only bottom friction shows an even more significant

energy contribution. This behaviour is explained by a strong nonlinear character in the velocity field on the background of nonlinear bottom friction. The momentum advection in the equations works in the regions of high nonuniform velocities as a dissipation factor. For this reason, the basic experiment (FTM) has a lower energy contribution.



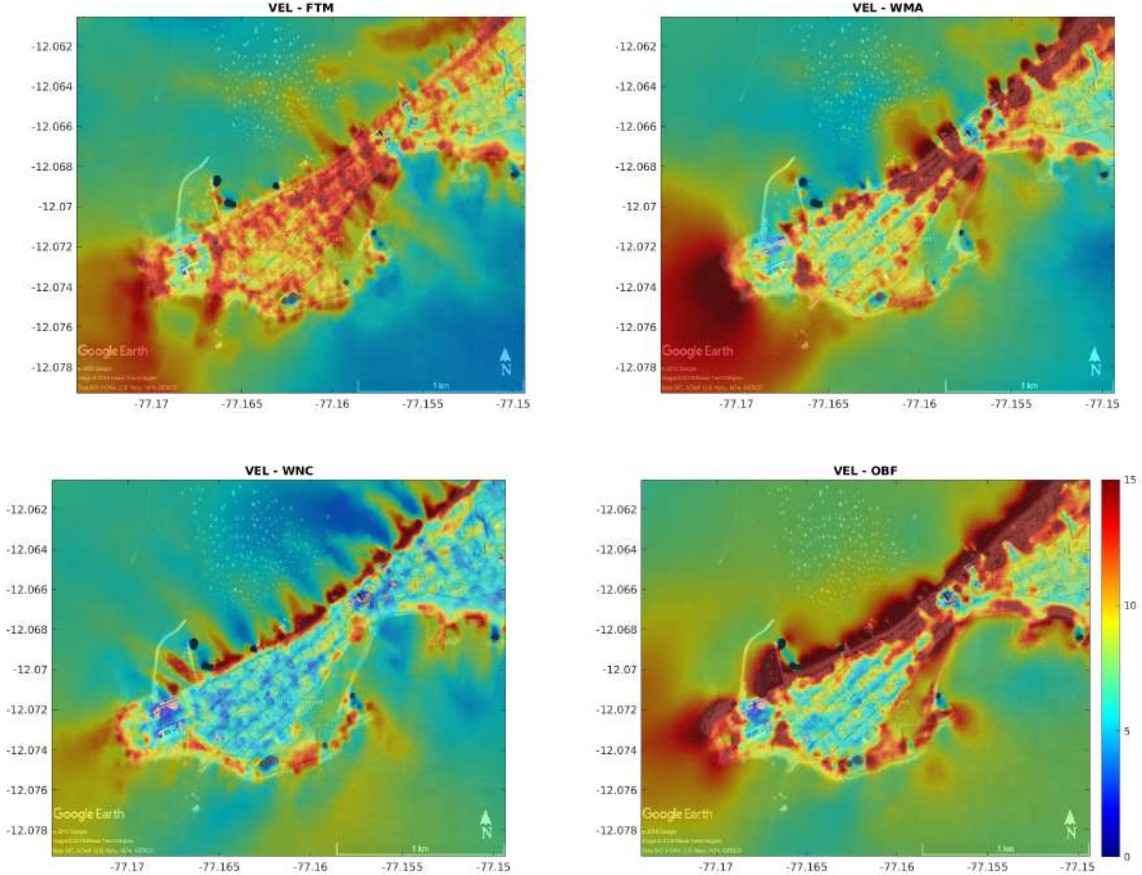

**Figure 12.** Maximum velocity modulus in [m/s] for the complete simulation period. Upper left panel - FTM; upper right - WMA; lower left - WNC; lower right - OBF. Basemap: © Google Earth 2018

As shown below, considering nonlinear terms does not increase or decrease the horizontal velocity or elevation in the flooded zone but completely changes the dynamic fields' structure.

We continue comparing the nonlinear effect by considering the dynamic characteristics: the maximum wave height and the velocity modulus in the Lima/Callao region. Figure 11 shows the spatial characteristics of the wave height for the four settings described above. As can be seen from the comparison, the level fields have significant differences both in amplitude and spatial structure. The wave height in the complete set has a local maximum in the narrowest part of the peninsula, with a wave height of up to 15 m. The WMA set-up calculations characterize the local maximum on the opposite side of the peninsula and more 270 significant amplitudes in the coastline zone. WNC calculations show underestimated tsunami wave amplitudes in the flood zone, with local maxima only along the coastline.

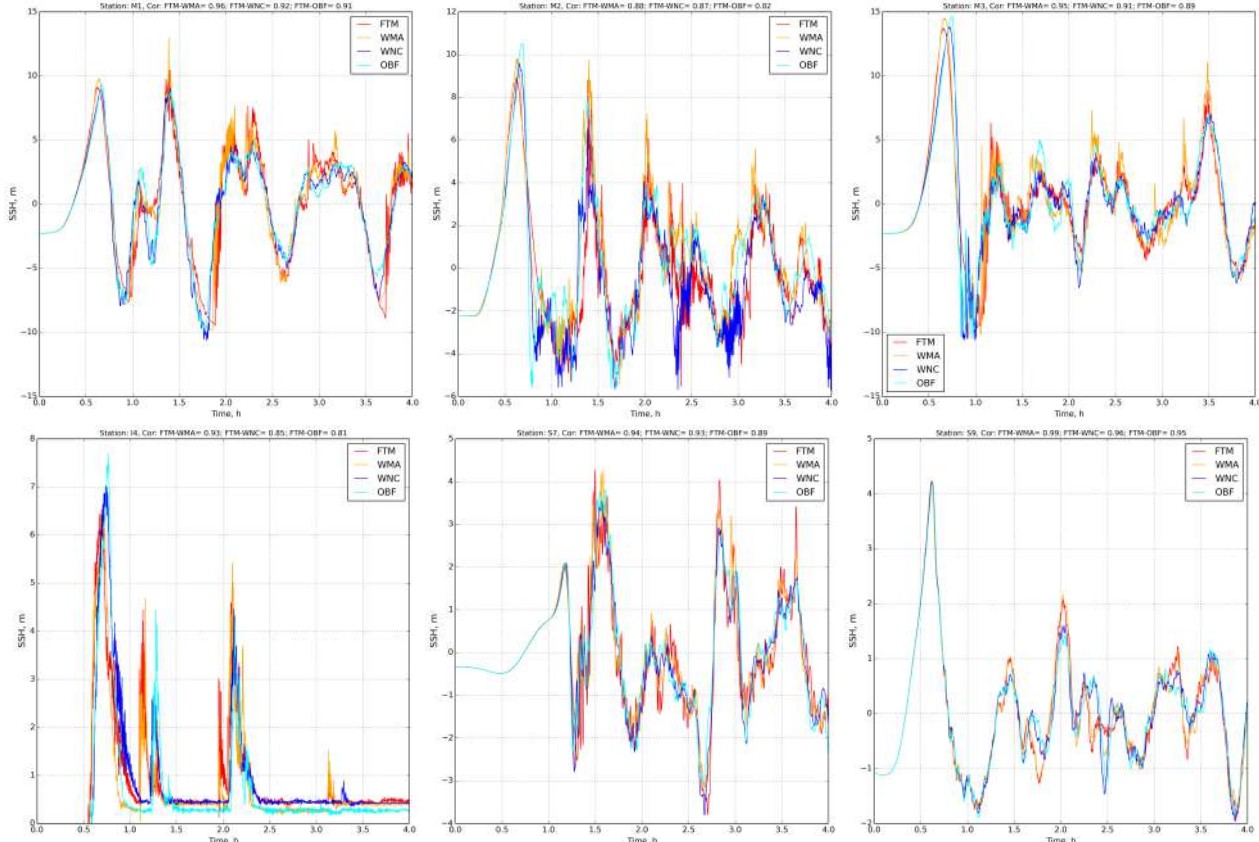

**Figure 13.** Time series of the sea surface elevation for the six points ($M1, M2, M3, I4, S7$ and $S9$) indicated in Fig. 2

A remarkable feature is highlighted by analysing the maximum horizontal velocity modulus in the same region. Figure 12 shows such fields for four settings. A characteristic feature of these fields is that in the absence of nonlinearity in the equation of momentum or continuity, there is a substantial intensification of the velocity along the coastline. The spatial structure of

such filaments is in good agreement with the bathymetric features of the coastal zone (see Fig. 2). In the case of a complete task, the horizontal velocities are more uniform and have maxima only in front of building structures, where the level gradient is maximum. In the absence of advection of the WMA experiment, the field structure is significantly uneven, with velocities significantly exceeding the local maxima of the FTM problem. In the absence of nonlinearity in the continuity equation WNC, the velocity amplitudes in the flood zone are greatly underestimated and have only local maxima along the coast. In the

absence of nonlinearity, except bottom friction (OBF experiment), the picture sums up the calculation without one or another nonlinearity - significantly intensifying alongshore currents and increasing velocities locally in the inundation area.

Figure 13 presents the results of comparing the wave height at stations for four problem settings - FTM, WMA, WNC, OBF. The positions of three stations (top panel) match those given in Jimenez et al.: Callao ($M1$), La Punta ($M2$) and Costa



Verde ($M3$). We also present the results for one station in the inundated area and two near coastlines to analyse the effect of
nonlinearity (bottom panel).

Comparison of the wave heights shows good agreement in the amplitude of the first wave at points $M1$ and $M2$ with the
amplitude values in (Jimenez et al., 2013) and are about 10 m. At the same time, at station $M3$, the wave amplitude in our
implementation is about 6 m less, which can be explained by a slightly different initial disturbance and different topographic
and bathymetric databases. So the depth of our bathymetry is more than 10 m, while in the above article, it is only 4 m.

An analysis of the results reveals general patterns in studying nonlinear terms for stations $M1 - M3$ offshore and coastal
station $I4$. So, in the WNC experiments and OBF, the shape of the first incoming wave changes somewhat. In the absence of
nonlinearity in the continuity equation, the waveform becomes flatter, which leads to a delay of the maximum of the first wave.
At the same time, at stations located at a slightly small distance from the coast, $S7$ and $S9$, the effect of nonlinearity in the
continuity equation is almost imperceptible. A similar shift of the maximum was also observed in the potential energy in the
flood zone (see Fig. 10). It can be assumed that calculating a tsunami wave without nonlinearity in the equation 2 changes the
wavefront in the entire simulated area. In addition, the absence of this kind of nonlinearity significantly reduces high-frequency
disturbances. The waveform during the whole simulation period becomes smoother at coastal points. The lack of advection in
the momentum equation 1, on the contrary, causes strong instability in the low-frequency spectrum, which once again confirms
that momentum advection acts as a dissipation factor in this case.

## 6   Conclusions

The results of modelling a tsunami wave along the coast of Peru for the 1746 scenario of two numerical models, Tsunami-
HySEA and TsunAWI, are analyzed. For both models, it was found that there is a slight phase shift in the wave propagation
velocity. Such a shift begins to manifest as the distance from the source of the tsunami wave increases. In the Tsunami-HySEA
model, the leading edge propagation velocity slightly lags behind the wave velocity in implementing the TsunAWI model.
We attribute this to the difference in dispersion errors in models with different spatial implementations. In the frequency
spectrum, the wave maxima are redistributed at the main frequency of the tsunami wave for this event (~1 hour) and at nearby
frequencies. Another difference between the two implementations is the more oscillatory nature of the dynamic characteristics
of the TsunAWI model compared to Tsunami-HySEA. This is easily explained by the fact that limiters in terms of the advection
moment were used in the last model, which stabilize the flow. With an increase in the coefficient of bottom friction, the
difference between the solutions of the two models practically disappears, which is confirmed by comparing the area and
volume of water masses in the flood zone (see Appendix). Thus, at the minimum bottom friction coefficient (Manning parameter
0.015) used in calculations, the flooded area in the TsunAWI model is 28.6 km$^2$, while in the TsunAWI model, it is 5% larger.
The volume of water masses in the TsunAWI model exceeds the Tsunami-HySEA results by about 42%. With an increase in
the friction factor to a value of $n = 0.045$, the difference in the flooded area decreases to 2.5%, and the difference in volume
drops to 13%. Otherwise, the results of the two models have a high level of similarity.



Accounting for the nonlinear terms of the shallow water equations is numerically complex enough that they are often neglected in models designed to generate warning products remarkably quickly, such as in an early warning system. These terms are relatively insignificant in the deep ocean, and it may become acceptable to neglect them in computations. On the contrary, the contribution of nonlinearity becomes very significant when the tsunami wave reaches the coast and plays a very important role, especially in flooding.

In this work, a detailed assessment of the influence of nonlinearity on the behaviour of the solution in the coastal and flood zones has been carried out. The shallow water equations are considered in four formulations - complete equations (FTM), equations without momentum advection in horizontal velocity equations (WMA), in the absence of nonlinearity in the continuity equation, when velocity divergence is considered without taking into account free surface perturbations (WNC) and in the presence of only nonlinearity in bottom friction terms (OBF).

A preliminary assessment of the bathymetry of the area showed that the sharp bottom slope is located a considerable distance from the coastline. The shelf zone extends for several kilometres behind it. It could be expected that, in this case, the nonlinearity in the continuity equation would be the maximum difference in the solution. A comparison of the results of wave height measurements at stations along the coast showed that the WNC experiment slightly shifts the beginning of the maximum, making the incoming tsunami wave flatter and, at the same time, practically does not change the wave amplitude. The lack of impulse advection (WMA) introduces the most significant changes in the amplitude of the incoming wave. Apparently, this is due to the orientation of the initial momentum of the free surface perturbation, which initially causes a significant shift of the velocity fields in space with a rather complex configuration of the coastline in the study area. We also attribute a significant increase, almost twofold, in the wave amplitude at coastal stations in the WMA experiment to a decrease in the dissipation of the solution, the role of which is partially played by the momentum advection term.

An analysis of the spatial solution in the flood zone without one or another nonlinearity introduces cardinal differences between the level and velocity fields from the complete problem statement. In the WNC experiment, the maximum wave amplitude on the coast is significantly underestimated, while in the WMA calculation, the flood maxima are overestimated.

Overall, we confirm that nonlinearity plays a decisive role in estimating flood areas, wave heights, current speeds and the spatial structure of flood maps. These factors should be considered when conducting numerical simulations of tsunami hazards to ensure that the solution persists in the nearshore and flooded zones.

## Appendix A: More results on the model comparison

### A1 Inundation properties

For the sake of completeness of the analysis, we will summarise some more results regarding the model comparison in this Appendix. The inundation area for a given Manning value is quite similar for the two numerical models. This is consistent with a similar assessment conducted for the Chilean cities Valparaíso, Viña del Mar and Coquimbo (Harig et al., 2022). Nevertheless, the spatial height distribution varies considerably locally. An example for n=0.020 is shown in Figs. A1 and A2. Especially





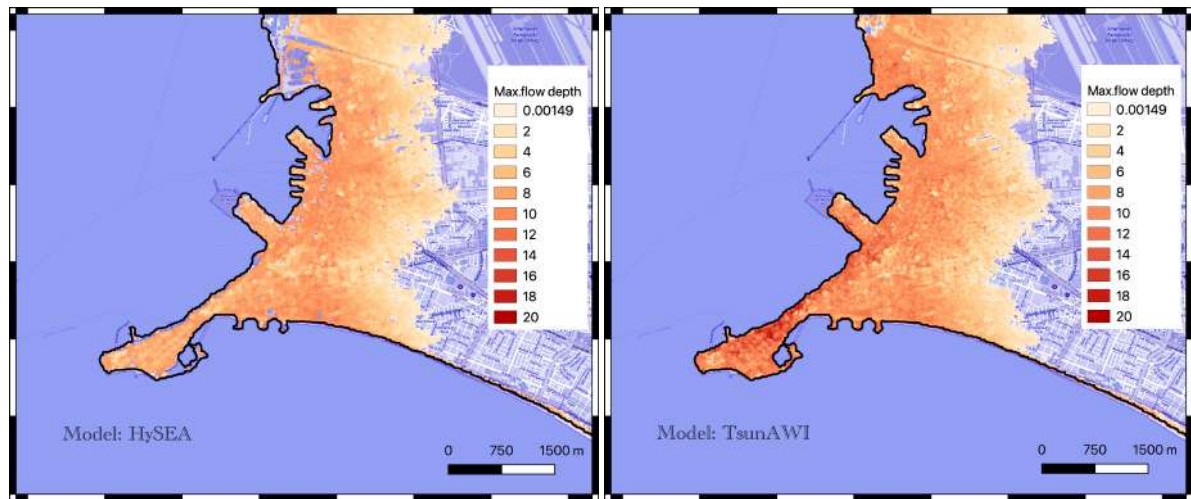

**Figure A1.** Maximum flow depth (in [m]) in La Punta and the Callao port area for simulations of four hours propagation in Tsunami-HySEA (left panel) and TsunAWI (right panel). Basemap: © OpenStreetMap contributors 2021. Distributed under the Open Data Commons Open Database License (ODbL) v1.0.

in the area close to the coastline, TsunAWI results in larger wave heights than Tsunami-HySEA, with even more significant differences for small Manning values. This is exemplified in the intersect comparison shown in the lower panel of Fig. A2.

350     A collection of all inundation polygons is shown in Fig. A3. The variation of inundation distance to the coastline may vary considerably depending on the bottom relief but is quite consistent for the two models.

    Table A1 summarizes some major inundation properties for both models and all Manning values as shown in Fig. 7. Here a minimum flow depth of 1cm defines the inundation extent. The corresponding values for a minimum flow depth of 1m are listed in Table A2. The Tables A1 and A2 also include the relative reduction of the quantities over the full Manning value range, 355   and we observe an inundation area reduction between 20% and 30% for the two models, with TsunAWI obtaining generally larger values (both in estimates and reductions). The largest differences between the models occur in the volume estimates. Here TsunAWI determines considerably larger values probably due to larger fluctuations and larger wave heights determined by this model close to the coast (see also Fig. A2). This difference is also visible in the medians of the maximum wave height.

    The velocity (modulus) distributions, together with the maximum values in the intersect, are shown in Fig. A4. The general 360   structure is very similar in the near shore range as well as on land, with regional variations. Peak values are somewhat larger for Tsunami-HySEA.

    Finally, we summarize inundation properties for the nonlinearity investigation in Fig A5 and Table A3. The figure shows the varying extent of the inundation area due to changes in the local velocity fields. The table highlights the different structures of the inundation process for the various setups. The simulation without momentum advection yields in the largest inundation area 365   and median of maximum flow depth also expressed in the largest potential energy obtained in the upper right panel of Fig. 10. In the same figure, it becomes clear that the experiment keeping only the bottom friction (OBF) obtains a larger maximum



**Table A1.** Properties of the inundation area obtained with both models for the full range of Manning values. The numbers specify the inundation area (with a minimum flow depth of 1 cm) in the bounding box of Fig. 6, the volume obtained by integrating the maximum flow depth over the whole area and the median of the maximum flow depth in the whole inundation area. The last row contains the relative reduction of all quantities over the Manning range as shown in Figure 7.

| Model | HySEA | | | TsunAWI | | |
|---|---|---|---|---|---|---|
| Manning n | Area [km$^2$] | Volume [Mio. m$^3$] | Median of max. flow depth [m] | Area [km$^2$] | Volume [Mio. m$^3$] | Median of max. flow depth [m] |
| 0.015 | 28.60 | 137.3 | 5.53 | 30.03 | 195.4 | 6.75 |
| 0.020 | 27.50 | 129.8 | 5.41 | 29.39 | 180.2 | 6.40 |
| 0.025 | 26.75 | 123.2 | 5.29 | 28.24 | 159.6 | 5.89 |
| 0.035 | 25.44 | 110.8 | 5.11 | 26.49 | 132.6 | 5.25 |
| 0.045 | 23.95 | 99.5 | 4.90 | 24.56 | 112.8 | 4.83 |
| 0.060 | 22.04 | 83.5 | 4.51 | 21.85 | 89.5 | 4.27 |
| rel. drop % | 22.9 | 39.8 | 18.1 | 27.2 | 54.2 | 36.7 |

**Table A2.** Inundation quantities just like in Table A1, however, obtained with a minimum flow depth threshold of 1 m.

| Model | HySEA | | | TsunAWI | | |
|---|---|---|---|---|---|---|
| Manning n | Area [km$^2$] | Volume [Mio. m$^3$] | Median of max. flow depth [m] | Area [km$^2$] | Volume [Mio. m$^3$] | Median of max. flow depth [m] |
| 0.020 | 26.6 | 133.3 | 5.91 | 28.3 | 192.6 | 7.16 |
| 0.020 | 25.7 | 125.9 | 5.79 | 27.5 | 177.2 | 6.82 |
| 0.025 | 25.0 | 119.3 | 5.69 | 26.2 | 156.5 | 6.32 |
| 0.035 | 23.6 | 107.3 | 5.47 | 24.4 | 129.5 | 5.70 |
| 0.045 | 22.3 | 96.1 | 5.23 | 22.6 | 109.7 | 5.25 |
| 0.060 | 20.5 | 80.2 | 4.84 | 20.0 | 86.5 | 4.66 |
| rel. drop % | 22.9 | 39.2 | 18.4 | 29.3 | 55.1 | 34.9 |




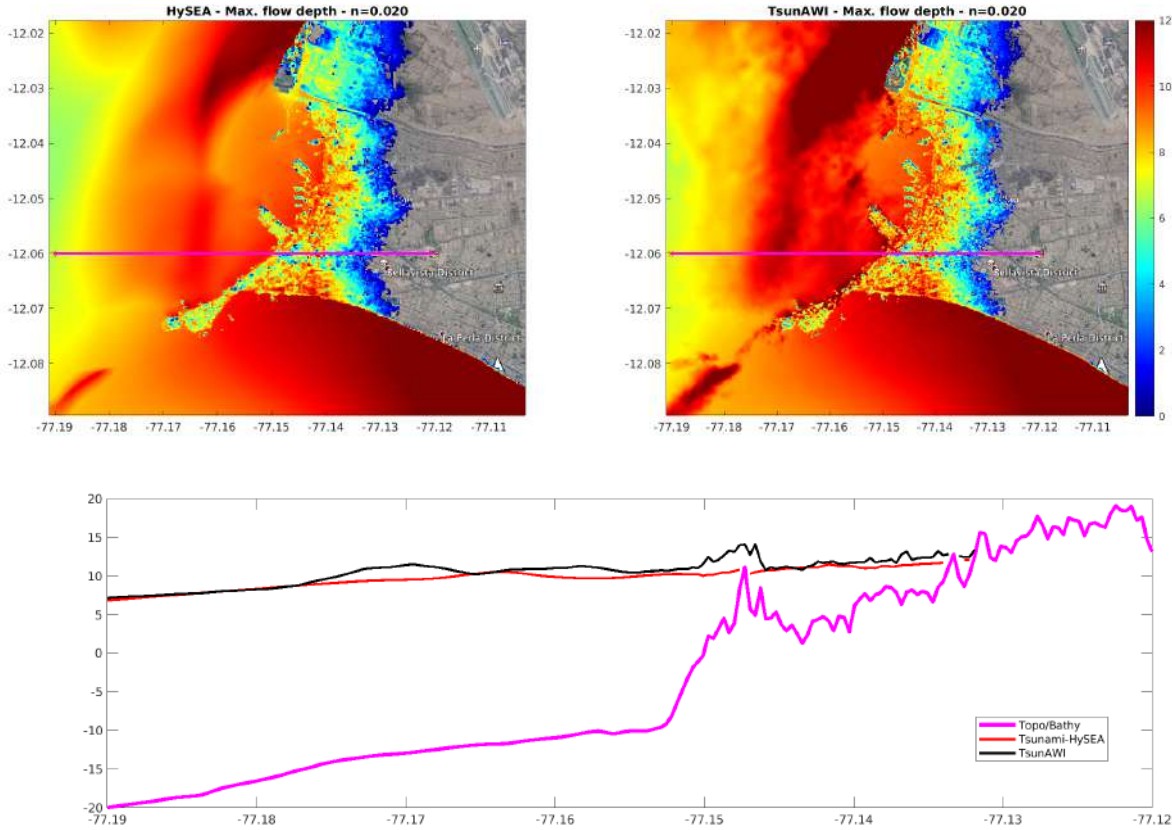

**Figure A2.** Upper panels: Maximum wave amplitude and flow depth (in m) in the Callao port area. Lower panel: Maximum wave amplitude projected to the intersect shown in the upper panels. All results obtained for Manning value $n = 0.020$. Basemap: © Google Earth 2018.

**Table A3.** Properties of the inundation area obtained for different setups with regard to nonlinear terms. Refer to Fig. A5. The values specify the inundation area (with more than 1cm flow depth) in the bounding box shown in Fig. 6, the volume obtained by integrating the max. flow depth over the whole inundation area and the median of the max. flow depth over the inundation area.

| Model | TsunAWI (Manning $n = 0.015$) | | |
|---|---|---|---|
| Experiment | Area [km$^2$] | Volume [Mio. m$^3$] | Median of flow depth [m] |
| FTM | 30.03 | 195.4 | 6.75 |
| OBF | 33.00 | 206.2 | 6.55 |
| WMA | 35.80 | 281.8 | 8.19 |
| WNC | 28.48 | 151.7 | 5.58 |

value of potential energy than the full model (FTM), however over the course of time, these experiments swap roles, and it is expressed, in the fact that OBF reaches a smaller median of maximum flow depth but a larger inundation area.



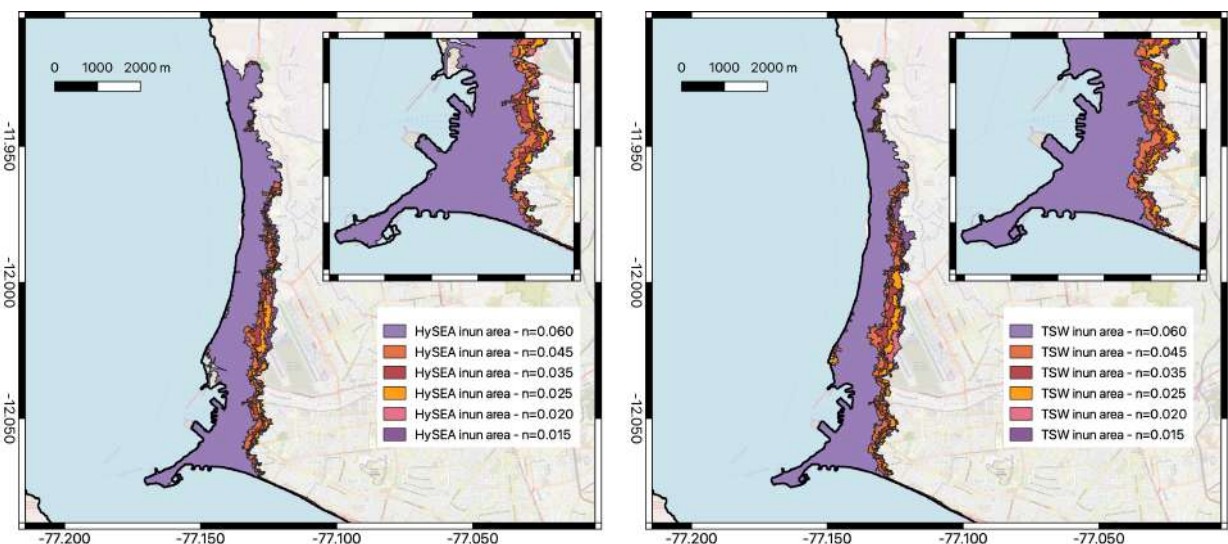

**Figure A3.** Inundation area with minimum flow depth of 1 cm for all Manning values obtained with Tsunami-HySEA (left panel) and TsunAWI (right panel). Basemap: © OpenStreetMap contributors 2021. Distributed under the Open Data Commons Open Database License (ODbL) v1.0.



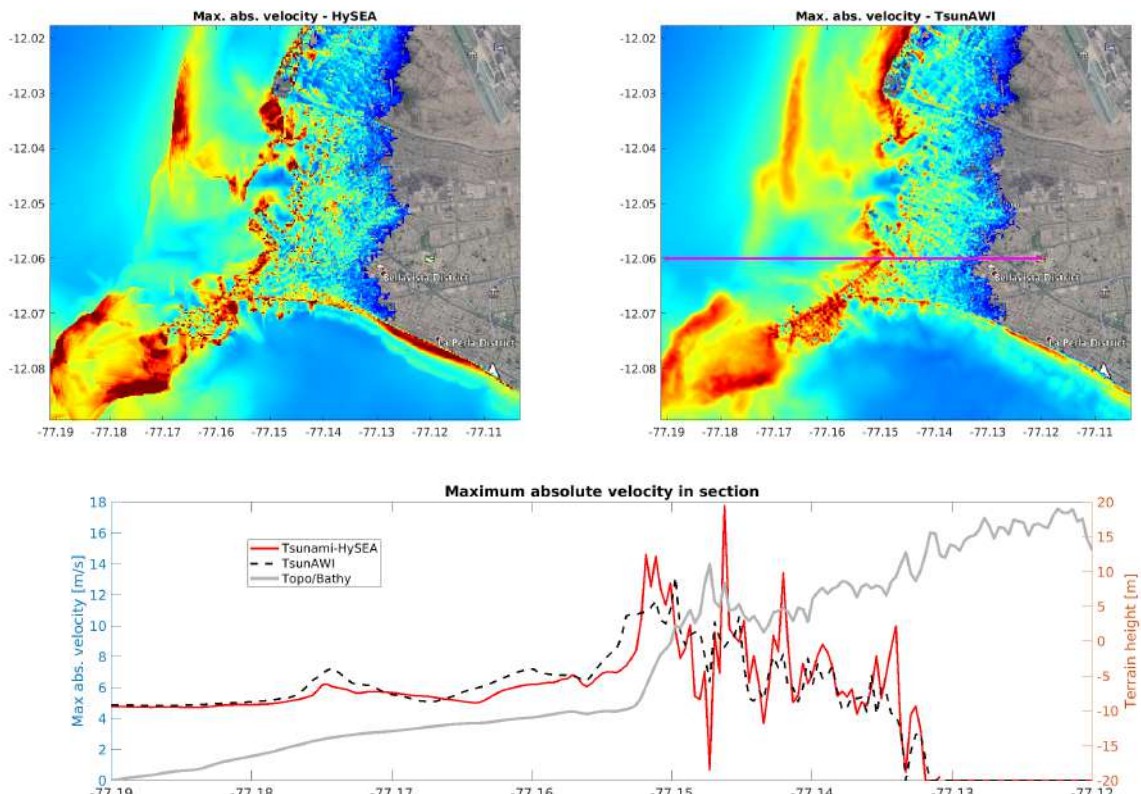

**Figure A4.** Upper panels: Maximum absolute velocity (in m/s) in the Callao port area obtained with Tsunami-HySEA (upper left) and TsunAWI (upper right). Manning value in both models is $n = 0.020$. Lower panel: Maximum absolute velocity projected to the intersection shown in the upper right panel. As a reference, the bottom relief is overlaid in grey colour. Basemap in upper panels: © Google Earth 2018.



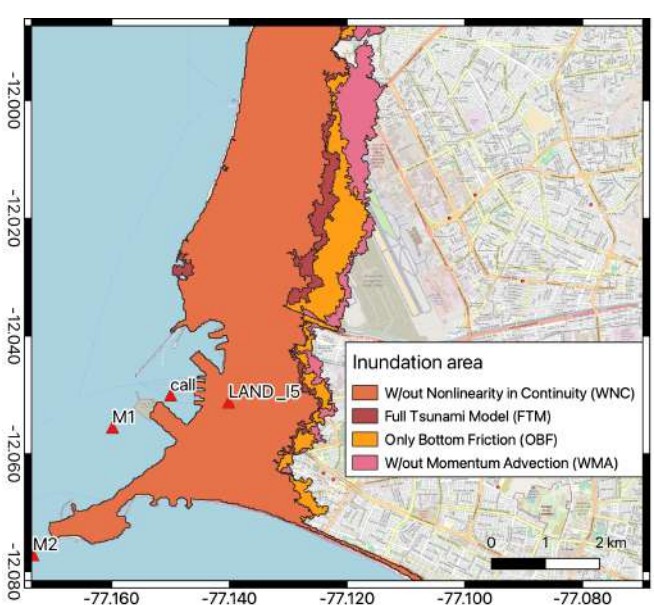

**Figure A5.** Inundation area obtained with TsunAWI for the experiments with the deactivation of different nonlinear terms. Basemap: © OpenStreetMap contributors 2021. Distributed under the Open Data Commons Open Database License (ODbL) v1.0.



*Author contributions.* Conceptualization, A.A. S.H and N.Z.; numerical model development, S.H, N.Z., A.A. and N.R.; numerical simula-
370 tions, S.H., A.A. and N.Z.; data processing, N.Z., S.H. and K.K; writing the original draft, A.A. S.H. and N.Z.; review and editing of the
manuscript, A.A., S.H., N.Z., N.R. and K.K.; visualization, S.H., N.Z. and A.A. All authors have read and agreed to the published version of
the manuscript.

*Competing interests.* The authors declare that they have no conflict of interest.

*Data availability.* The data presented in this study are available on request from the corresponding author.

375 *Acknowledgements.* Part of this research was funded by the German Federal Ministry of Education and Research within the projects RIES-
GOS and RIESGOS2.0 (grant numbers 03G0876C and 03G0905C). NZ was funded by the Marie Skłodowska-Curie grant agreement
H2020-MSCA-COFUND-2016-75443 and the ChEESE-2p. We acknowledge the Edanya Group at University of Málaga for sharing the
Tsunami-HySEA software. We are grateful with J. Macías and C. Sanchez for their support with the Tsunami-HySEA code. Most figures
were generated with QGIS (QGIS Development Team, 2009), Matlab Inc. (2022) and Global Mapping Tool Wessel et al. (2019).





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
