# Peer review of "Nonlinear processes in tsunami simulations for the Peruvian coast with focus on Lima/Callao"

_EGUsphere, 2023_

## Referee Comment (RC2)

**Review**

On the manuscript **"Nonlinear processes in tsunami simulations for the Peruvian coast with focus on Lima/Callao"** by Alexey Androsov, Sven Harig, Natalia Zamora, Kim Knauer, and Natalja Rakowsky.

**Overview: Conditional Acceptance Upon Minor Revision**
* * *
The paper presents numerical simulations of tsunami flooding in Lima and Callao caused by a historical 9.0 magnitude earthquake on the Peruvian coast including strong nonlinear terms. Two numerical codes are implemented for solving the shallow water equations, although with different spatial approximations. The study is mainly concerned with the properties of momentum advection, bottom friction, and volume conservation. The text is mostly well-written and provides an important contribution to the study of ancient natural hazards. However, below you will find minor specific comments (including the references therein) that should be addressed/added towards the publication of this work:

1. The introduction could be better formulated and widened for a broad spectrum of readers. The authors do not discuss how inhomogeneous coastal and geomorphological processes can affect tsunami run-up heights and arrival times estimation from simulations in the introduction. Note that although this is related to the first type of tsunami modeling (using the authors' own jargon), indeed, the Sumatra 2004 earthquake had a similar magnitude and demonstrated that complexities in large tsunamis are not well understood (Arcas and Titov, 2006; Broutman et al., 2014; Rabinovich et al., 2017), highlighting the need to understand better and estimate local bathymetry. Although nonlinear terms in the tsunami code is one of the signatures of this work, the authors also do not discuss the disturbance on tsunami signals by local bathymetry and coastal topography, resonance, refraction, and nonlinearity (Mofjeld, 2009) concisely (only the reflection is discussed later at the end of the introduction). It would be good to have a short discussion on the commonality of inhomogeneous effects affecting water wave hazards (Shuto, 1967; Tuck and Hwang, 1972; Pelinovsky and Mazova, 1992; Golinko et al., 2006; Tsai et al., 2013; Chugunov et al., 2020; Mendes et al., 2022; Mendes and Kasparian, 2022). Finally, both introduction an conclusions should mention that wave run-up, and by consequence flooding, is stochastic and assumes a statistical distribution over spatial domains regardless of the chosen deterministic numerical model, see for instance (Choi et al., 2002, 2006; Geist and Parsons, 2008).

2. Although the authors do plot bathymetry charts, they do not give a sense of slope magnitudes. I recommend they to briefly describe in the text the mean slope magnitude and maximum as well. Not only water depth is relevant, but how fast this change take place as well, i.e. the slope (see references in the comment above).

3. In section 5 the authors discuss nonlinearity. I believe it to be important to show how the three types of nonlinear forcings vary spatially so that a proper evaluation of their effect is compared with tsunami run-up and flooding characteristics. The authors are already similar to that for a sample time series and spatial distribution of the cumulative effect of the three nonlinear forcings or by omitting one of them, but a separate analysis of each is also called for. The present plots on the effects of omitting one of the nonlinear forcings analyze only the outcome, but not the nonlinear term itself varying spatially.

4. The quality of figures is pretty low and they are generally small. Provided there is no page limit, I would support larger figures with higher resolution whenever possible.

5. I see no reason for the context of appendix A to be detached from the text itself, I recommend bringing it back.

**Conclusion**

The reviewer thanks for the opportunity to read this important work. Overall, I support the publication of this preprint once all these minor issues have been clarified/amended.

**References**

Arcas, D., Titov, V., 2006. Sumatra tsunami: lessons from modeling. Surveys in Geophysics 27, 679–705.

Broutman, D., Eckermann, S.D., Drob, D.P., 2014. The partial reflection of tsunami-generated gravity waves. Journal of the Atmospheric Sciences 71, 3416–3426.

Choi, B.H., Hong, S.J., Pelinovsky, E., 2006. Distribution of runup heights of the december 26, 2004 tsunami in the indian ocean. Geophysical research letters 33.

Choi, B.H., Pelinovsky, E., Ryabov, I., Hong, S.J., 2002. Distribution functions of tsunami wave heights. Natural Hazards 25, 1–21.

Chugunov, V.A., Fomin, S.A., Noland, W., Sagdiev, B.R., 2020. Tsunami runup on a sloping beach. Computational and Mathematical Methods 2, e1081.

Geist, E.L., Parsons, T., 2008. Distribution of tsunami interevent times. Geophysical Research Letters 35.

Golinko, V., Osipenko, N., Pelinovsky, E., Zahibo, N., 2006. Tsunami wave runup on coasts of narrow bays. International Journal of Fluid Mechanics Research 33.

Mendes, S., Kasparian, J., 2022. Saturation of rogue wave amplification over steep shoals. Phys. Rev. E 106, 065101.

Mendes, S., Scotti, A., Brunetti, M., Kasparian, J., 2022. Non-homogeneous model of rogue wave probability evolution over a shoal. J. Fluid Mech. 939, A25.

Mofjeld, H.O., 2009. Tsunami measurements. In A. Robinson E. Bernard (Eds.), The sea, Vol.15 , 201–235.

Pelinovsky, E., Mazova, R.K., 1992. Exact analytical solutions of nonlinear problems of tsunami wave run-up on slopes with different profiles. Natural Hazards 6, 227–249.

Rabinovich, A.B., Titov, V.V., Moore, C.W., Eblé, M.C., 2017. The 2004 sumatra tsunami in the southeastern pacific ocean: New global insight from observations and modeling. Journal of Geophysical Research: Oceans 122, 7992–8019.

Shuto, N., 1967. Run-up of long waves on a sloping beach. Coastal Engineering in Japan 10, 23–38.

Tsai, V.C., Ampuero, J.P., Kanamori, H., Stevenson, D.J., 2013. Estimating the effect of earth elasticity and variable water density on tsunami speeds. Geophysical Research Letters 40, 492–496.

Tuck, E., Hwang, L.S., 1972. Long wave generation on a sloping beach. Journal of Fluid Mechanics 51, 449–461.

---

## Author Comment (AC1)

**Replies to the reviewer's comments and suggestions.**

1. Clarity of experiment design. It is not clear to me how the four nonlinear experiments were designed, especially the fourth one (OBF). From what I understand, the first experiment is to include all the nonlinear terms in the governing equations (FTM) which forms a base model (control) for the study; the second experiment only removes advection terms in the momentum equations; the third expeirments only removes free surface elevation out of the divergence term in the continuity (mass conservation) equation; and then the fourth experiment would be only removing the free surface component out of H in the bottom friction formula. This means as each comparative experiment among experiments 2-4 investigates effects of a single nonlinear term while keeping the other two. This makes perfect sense as it would tell us the effects/contribution of that nonlinear term when comparing its result with FTM outputs. However, the description on the fourth expeirment (OBF) on page 13 seems telling a different story - it only keeps bottom friction term while removing the other two nonlinear terms (i.e. advection term in momentum equations and free surface elevation in the divergence term in the mass equation). Could you please revise section 5 a bit to clearly state how the experiments were designed?

   > As you noticed, nonlinearity analysis experiments are designed so that FTM is a control calculation with all the system's nonlinearity. WMA calculation without taking into account moment advection in the equation of motion. In this case, two other elements of nonlinearity are present. WNC without calculating the free surface in the mass conservation equation, while moment advection and bottom friction are present in the system. Final experiment: turning off all nonlinearity except for bottom friction. Bottom friction is the only stabilizer of the numerical solution, and calculations without considering it are only possible in

*some test problems, for example, the roll-up of a symmetrical wave onto a flat shore. In this case, the wave velocities along the entire front are quasi-uniform, and the scheme is stabilized due to weak internal viscosity. In the real modelling presented in the work, complete elimination of bottom friction is not possible due to the strong heterogeneity of the solution in the flood zone. A more or less stable solution is achieved by a significant (by orders of magnitude) reduction in the time step, which in turn leads to an imbalance of the time derivative and gradient terms, i.e. the change in speed over time can be calculated with significant errors due to the smallness of their difference, which will lead to instability of the solution. In this regard, to evaluate the effect of the bottom friction coefficient, we additionally conduct a series of experiments with its different values and monitor the nature of the solution.*

*A necessary clarification has been included in the text.*

2. Figure improvements

- Colour scheme for figure 1 is not ideal, a bit too dark. I recommend to use the same colour scheme as figure 2's for clarity and consistency.
-Figure 2: panel label a) and b) are missing.

-Figure 11, Figure 12, Figure A2, Figure A4: could you please add coastal line contours to assist with data interpretation?

*The mentioned figures were updated, the coastline was added as a topography contour line and font sizes were somewhat enlarged to facilitate reading.*

2. Appendix

- Figure A4 shows shows large descrepancies in maximum absolute veocities along the section between Tsunami-HySEA and TsunAWI simulations. Could you please provide some

comments/discussions on what might be the factors contributing to the differences?

> *The difference in absolute velocity is not so significant, in our opinion. The absolute maxima are well consistent along the section, and the difference in amplitudes is mainly due to the different spatial resolution of the two models in the flood zone, where the Tsunami-HySEA has a spatial resolution of 12 meters, and the other is slightly worse for this particular region. And a model with better resolution naturally describes extrema better. But note that the Tsunami-HySEA has fine resolution only in a specific area, and for the rest of the domain, the resolution is relatively coarse. At the same time, TsunAWI has quasi-uniform high resolution for the entire coastal zone.*

For the two upper panel figures, their titles are not quite right. What the colour scale shows are maximum wave amplitudes in water area and maximum flow depth on land; but the figure caption describes this correctly.

> *Thank you for pointing us at the discrepancy. We updated the titles and added a colorbar to figure A4 (now A5).*

*We thank the reviewer for carefully reading the article and useful comments.*

*Authors.*

---

## Author Comment (AC2)

**Replies to the reviewer's comments and suggestions.**

1. The introduction could be better formulated and widened for a broad spectrum of readers.

   *If the reviewer does not object, we will try to divide the response to the first paragraph of comments into several subparagraphs to respond to suggestions and comments more thoroughly. Changes in the text of the article after editing are highlighted in red.*

The authors do not discuss how inhomogeneous coastal and geomorphological processes can affect tsunami run-up heights and arrival times estimation from simulations in the introduction. Note that although this is related to the first type of tsunami modeling (using the authors' own jargon), indeed, the Sumatra 2004 earthquake had a similar magnitude and demonstrated that complexities in large tsunamis are not well understood (Arcas and Titov, 2006; Broutman et al., 2014; Rabinovich et al., 2017), highlighting the need to understand better and estimate local bathymetry. Although nonlinear terms in the tsunami code is one of the signatures of this work, the authors also do not discuss the disturbance on tsunami signals by local bathymetry and coastal topography resonance, refraction, and nonlinearity (Mofjeld, 2009) concisely (only the reflection is discussed later at the end of the introduction).

   *In the introduction, we have discussed the lack of information in tsunami wave modelling about bathymetry/topography and its effect on tsunami wave heights. At the same time, we would like to emphasise that in this study, we do not experience problems with a high-quality database of bathymetry and topography. The database used has a resolution of 300m for bathymetry and 12m for the dry zone (our mesh resolution in Tsunami-HySEA model).*

*And the task before us was not to assess the quality of the topography. A description of the database used is given in paragraph 3.2.*

*The effect of resonance and refraction is essential in studying tsunami waves. We have added a phrase to the introduction about the importance of this phenomenon (with reference). We hope that in the near future, we will try to return to this issue in our research using the example of the coast of Chile, which is the most interesting object for studying this process.*

It would be good to have a short discussion on the commonality of inhomogeneous effects affecting water wave hazards (Shuto, 1967; Tuck and Hwang, 1972; Pelinovsky and Mazova, 1992; Golinko et al., 2006; Tsai et al., 2013; Chugunov et al., 2020; Mendes et al., 2022; Mendes and Kasparian, 2022).

*Additional material has been added to the introduction.*

*"Megatsunamis, like the ones that occurred in the Indian Ocean in 2004 and after the Tohoku earthquake in 2011, are distinguished by their impacts over significant distances. Both of these massive tsunamis were recorded along the South American coasts \citep{Rabinovich2017}. As the tsunami flowed, waves suffered reflection from continents and from the abrupt changes in the bathymetry and coastal topography \citep{Arcas2006, Rabinovich2017} and were affected by various atmospheric processes \citep{Broutman2014}, which significantly distorted the original signal. As the tsunami waves approached the coast, they were modified considerably by the continental shelf and local topography. In addition to the factors influencing the propagation of a tsunami wave described above, it was also noted that tsunami resonance and associated fluctuations in shelf and bay modes could play a crucial role in amplifying tsunami waves \citep{aranguizetal2019}. All these processes*

*are highly nonlinear and can significantly depend on the quality of bathymetry and topography data e.g. \citep{Sepulveda2020"*

Finally, both introduction an conclusions should mention that wave run-up, and by consequence flooding, is stochastic and assumes a statistical distribution over spatial domains regardless of the chosen deterministic numerical model, see for instance (Choi et al., 2002, 2006; Geist and Parsons, 2008).

*Recommended additions have been made to the introduction*

*"The theory of the run-up of long waves onto the shore is of considerable interest. This problem of various long symmetrical or antisymmetrical waves with the same steepness of the front and rear slopes of unbroken waves onto a flat slope is quite well developed from a mathematical point of view within the framework of the nonlinear theory of shallow water, allowing for an analytical solution \citep{Shuto1967, Tuck1972, Pelinovsky1992, masselpelinovsky2001, tintitonini2005}. As a result, formulas for run-up height can be parameterized to include the height and length of the suitable wave and the distance to shore. The numerical results of such models are reasonably in agreement with laboratory experiments. At the same time, according to numerous observations of the 2004 tsunami, due to the transformation of the wave when moving along heterogeneous bathymetry, a strongly deformed wave with a noticeable steepness of the front slope approaches the shore. \cite{didenkulovaetal2006} work showed that a wave with an increased steepness of the front slope penetrates further onto the coast than a wave with a symmetrical profile."*

*The authors are fully aware that only a probabilistic study can account for the uncertainties with regard to the source of any given tsunami event, and estimates of the resulting inundation process need to be studied in a probabilistic sense. In this study, however, we focus on the contributions of nonlinear terms and parameter settings of the shallow water equations and therefore, we fix the source and investigate the inundation as a highly*

*nonlinear but deterministic process. If there are more aspects the reviewer would like to address, please let us know.*

2. Although the authors do plot bathymetry charts, they do not give a sense of slope magnitudes. I recommend they to briefly describe in the text the mean slope magnitude and maximum as well. Not only water depth is relevant, but how fast this change take place as well, i.e. the slope (see references in the comment above).

> *Thanks for the recommendation. We have included a figure showing the bathymetry gradient in the area's study and added some description in the text in the appendix.*

3. In section 5 the authors discuss nonlinearity. I believe it to be important to show how the three types of nonlinear forcings vary spatially so that a proper evaluation of their effect is compared with tsunami run-up and flooding characteristics. The authors are already similar to that for a sample time series and spatial distribution of the cumulative effect of the three nonlinear forcings or by omitting one of them, but a separate analysis of each is also called for. The present plots on the effects of omitting one of the nonlinear forcings analyze only the outcome, but not the nonlinear term itself varying spatially.

> *In addition to the spatial influence of each nonlinearity term, an analysis of the contribution of one or another term is given. The conclusion of the article repeats the findings in sufficient detail. Thus, the text states that the effect of the absence of moment advection introduces the impact of a significant reduction in horizontal smoothing. In other words, moment advection for a highly inhomogeneous field in the flood zone acts as a factor stabilizing the solution. The absence of nonlinearity in the continuity equation introduces the effects of a sharp steepening of the front of the oncoming wave and its destruction upon reaching the coastline. This effect is especially evident in morphometric features in the coastal zone. Nonlinearity due to bottom friction is included in each calculation since this is*

*practically the only stabilizer of the numerical scheme. Without considering bottom friction, it is possible to carry out calculations only in a homogeneous (concerning the topography and bathymetry of the area) with a homogeneous wave.*

4. The quality of figures is pretty low and they are generally small. Provided there is no page limit, I would support larger figures with higher resolution whenever possible.

> *Done.*

5. I see no reason for the context of appendix A to be detached from the text itself, I recommend bringing it back.

> *We partially transferred the results concerning the study of nonlinearity (a table and figure showing the flood zone for our experiments) into the main part of the work. We still decided to leave the remaining results on comparing models in the appendix so as not to overload the main parts of the article.*

*We are grateful to the reviewer for his careful reading of the article, useful comments, and warm words of support for the results of the work.*

*Authors.*

---

## Author Response (AR2)

I'm pleased to accept your author for publication in our journal pending the following minor revisions which will be reviewed by the editor.

*We are grateful to the editor for his careful reading and useful comments.*

1.0    References. Please do a final check that all text AND all equations where you have facts, information, ideas that are not your own (or are yours but from another paper) are properly cited. Just as one example, Equations 1 and 2 have no in-text citation. Although general formulae, they still should have citations.

*Reference to shallow water equations added.*

2.0    Please ensure that all figure captions are complete enough to be self-standing, if the figure were to be copied apart from the text, and that all data/parts of the figures that are not your own have an appropriate citation.

*We looked at all the captions for the figures. Missing information has been added to some of the figures.*

3.0 Figures:

3.1 Please ensure that if you use degrees latitude and longitude, that you also put W, N, E, S as appropriate along the axes, to avoid any confusion;

3.2 In every location you have 'm' for elevation, I believe you mean m asl? If so, the asl needs to be added (and the first time, stated this means above sea level). This should be in legends, axis labels, and text of figure captions or text overall.

3.3 For latitude and longitude, please decide on the number of decimal places and stick to it. So 11.95, 12.00 (not 12). Please check this for axes, legends, etc.

3.4 Please check font size for all figures that it is not too small. For example (but please check all) text size in Figure 4 and 7 are much too small to be visible to the average reader.

3.5 Make sure units are in ALL of your legends (unless unitless). For example, Figure 6 has no units for max flow depth.

3.6 Figure 7. I suggest you not use 'mio.' as an abbreviation for 10^6. Just use 10^6. TS and HSW never defined in the figure caption.

3.7 For Figures such as Figure 10, I recommend you use something in addition to colour to distinguish these curves. In this case you could also use dot or dash dot, or thickness of the line. Colour alone can cause issues for the 8% of males and .5% women with colour blindness.

*All figures have been reviewed and modified according to your requirements. Added information on the axis. On some graphs (such as Fig. 10), the line thicknesses have been changed, i.e. each line has its own thickness.*

3.0 Please consider (not required, but will help out the reader) a table of variables used, and acronyms.

*We decided not to create an additional table with acronyms. All acronyms are described in the text of the article as they appear, and an attentive reader will not have a problem understanding their meaning.*

4.0 Self-plagiarism. I see that you have published on this topic before. That is fine, but be clear about text (you have a couple hundred words from two places you have published in before, almost word for word) that comes from another place, as if you were citing someone else. Related to this, please change 'chapter' to 'paper' in Line 273. You previously published a chapter but now are writing a paper.

*‚chapter' to ‚paper' – corrected.*

*We checked the article for plagiarism. The largest piece of borrowing is shown in the photograph. But each phrase borrowed from other publications is marked with a link. The description of the tsunami model is partially borrowed from our previous publications with some abbreviations. There are also new additions related to the filtering method. All references are provided. We guarantee that the text of the article is original.*

[Figure]

GO PRO

| | | | |
|---|---|---|---|
| Q | Ø | 🎧 | 🕑 |
| Profundo Búsqueda | Sin anuncios | Apoyo | Preciso Informes! |

Conviértete en profesional

type includes operational models (Titov et al., 2005b), whose main task is to estimate the time of arrival 30 of a tsunami wave and its height on a time scale faster than wave propagation in real-time. Often, such models are linearized and do not perform calculations in the inundation zone. The tsunami wave height metric can be obtained using the so-called "amplification factor", which describes the relationship between the wave height in the open sea and the maximum inundation height for waves with different wave characteristics (Glimsdal et al., 2019). Another approach to assessing the tsunami threat can be considered based on Green's summation, using the parameters of the seismic source (Miranda et al., 2014) as input. 35 Green's functions are calculated using a numerical SW model for linearized equations at points of most significant interest. Numerical models of the second kind should describe the inundation area with a very high spatial resolution and have reliable numerical wetting/drying schemes, which are associated with relatively high energy consumption in the computational aspect. In addition, often, the time between the occurrence of a tsunami and its approach to the coast is minimal, and then pre-calculated databases of possible scenarios of tsunami sources and numerical modelling (Macías et al., 2017; Rakowsky et al., 2013) come 40 to the rescue. The advantage of such models compared to operational ones is a more accurate and detailed description of the processes